# Interplay between Lipid Metabolism, Lipid Droplets, and DNA Virus Infections

**DOI:** 10.3390/cells11142224

**Published:** 2022-07-17

**Authors:** Mónica A. Farías, Benjamín Diethelm-Varela, Areli J. Navarro, Alexis M. Kalergis, Pablo A. González

**Affiliations:** 1Millennium Institute on Immunology and Immunotherapy, Departamento de Genética Molecular y Microbiología, Facultad de Ciencias Biológicas, Pontificia Universidad Católica de Chile, Santiago 8330025, Chile; mrfarias@uc.cl (M.A.F.); bmdiethelm@uc.cl (B.D.-V.); ahnavarro@uc.cl (A.J.N.); akalergis@bio.puc.cl (A.M.K.); 2Departamento de Endocrinología, Facultad de Medicina, Escuela de Medicina, Pontificia Universidad Católica de Chile, Santiago 8330025, Chile

**Keywords:** lipid metabolism, neutral lipids, lipid droplets, DNA viral infections

## Abstract

Lipid droplets (LDs) are cellular organelles rich in neutral lipids such as triglycerides and cholesterol esters that are coated by a phospholipid monolayer and associated proteins. LDs are known to play important roles in the storage and availability of lipids in the cell and to serve as a source of energy reserve for the cell. However, these structures have also been related to oxidative stress, reticular stress responses, and reduced antigen presentation to T cells. Importantly, LDs are also known to modulate viral infection by participating in virus replication and assembly. Here, we review and discuss the interplay between neutral lipid metabolism and LDs in the replication cycle of different DNA viruses, identifying potentially new molecular targets for the treatment of viral infections.

## 1. Introduction

Lipid droplets (LDs) are lipid-rich organelles that are mainly associated with lipid storage and the maintenance of lipid homeostasis in the cell. LDs are composed of neutral lipids, mainly triglycerides (TAGs) and cholesterol esters (CEs), but also diglycerides (DAGs), which altogether are enveloped in a monolayer of phospholipids with LD-associated proteins [1,2,3]. While class I proteins are mainly associated with the membrane of the LDs through hydrophobic hairpins, class II proteins usually bind to LDs through amphipathic α-helices [1,4].

LDs are mainly found in the cytoplasm, where they can interact with other organelles, such as the endoplasmic reticulum (ER), mitochondria, peroxisomes, and endosomes [1,5,6]. However, LDs have also been observed in the nucleus, where they have been associated with diverse roles such as lipid metabolism and genome regulation [7,8].

LDs are mainly described as a source of energy storage in the cell, a process that is carried out through fatty acid (FA) release from the breakdown of neutral lipids, which then interact with mitochondria or peroxisomes for β-oxidation [9,10,11]. However, other cellular roles have been associated with LDs, such as the maintenance of ER homeostasis, the regulation of autophagy, and cellular protection against lipotoxicity, among others [12,13,14]. Additionally, LDs are increasingly being described as immune modulators, and have been associated with inflammatory responses in leukocytes and macrophages [15,16,17,18]. Moreover, in the context of cancer, LDs have been reported to block antigen presentation by dendritic cells (DCs) to T cells, hampering the activation and proliferation of the latter [19,20].

On the other hand, viruses modulate numerous cellular processes to carry out their replication cycles. Viruses can alter cellular processes, such as cell replication, gene transcription, mRNA translation, and post-translational modifications [21,22,23,24,25]. Moreover, viral infections are known to modulate cytoskeleton remodeling and the secretory pathway to mobilize viral components within the cell and overall to promote virus assembly and export [26,27]. Additionally, cellular signaling pathways are also modulated by viruses, such as those related to phosphatidylinositol phosphate kinases (PIKs), mitogen-activated protein kinases, (MAPK) toll-like receptors (TLRs), and interferon (IFN)-related pathways, to name a few [28,29,30]. The function of cellular organelles and key cellular processes may also be altered by viral infections, such as the unfolded protein response (UPR), autophagy, apoptosis, mitochondrial function, heat stress, and hypoxia [31,32,33,34,35]. Finally, viral infections modulate important cellular metabolic pathways, such as glucose and lipid metabolism as well [36,37].

Here, we review and discuss the main processes involved in neutral lipid synthesis and degradation that impact LD formation. Moreover, we revise how DNA viral infections modulate neutral lipid metabolism and LDs, thoroughly analyzing how these processes and lipid structures impact the replication cycle of DNA viruses.

## 2. Lipid Droplet Metabolism

### 2.1. Lipid Droplet Formation

LD biogenesis is usually accompanied by neutral lipid synthesis, mainly TAGs and CEs (Figure 1). The de novo synthesis of TAGs occurs in four reactions catalyzed by the enzymes glycerol-3-phosphate O-acyltransferase (GPAT), 1-acylglycerol-3-phosphate O-acyltransferase (AGPAT), and phosphatidic acid phosphatase (PAP)/lipin, as well as enzymes belonging to the family diacylglycerol O-acyltransferases (DGAT) that generate the esterification of DAGs into TAGs [2]. TAG synthesis also occurs through the Kennedy pathway, which involves the de novo synthesis of glycerolipids through the combination of glycerolphosphate and fatty acid acyl-CoA to yield TAGs [38]. FA synthesis, in turn, is required for generating fatty acid acyl-CoA, a process first mediated by the enzyme acyl-CoA carboxylase (ACC), then fatty acid synthase (FAS), and finally requiring activation with acyl-CoA, a process catalyzed by the enzyme acyl-CoA synthetase (ACS) [2,39,40]. A second TAG synthesis pathway called the monoacylglycerol pathway is responsible for recycling monoacylglycerols (MAGs) and DAGs generated by the hydrolysis of TAGs, involving a re-esterification process [41]. However, independent of which synthesis pathway is used, the formation of TAGs will ultimately be catalyzed by either of the enzymes DGAT1 or DGAT2, both present in the ER [42,43].

On the other hand, the de novo synthesis of CE requires the synthesis of sterols to produce cholesterol molecules, which will undergo an esterification process that generates CE [44]. The synthesis of cholesterol is a complex process that involves more than 30 biochemical reactions and begins with the formation of 3-hydroxy-3-methylglutaryl-coenzyme A (HMG-CoA) from an acetyl-CoA molecule and acetoacetyl-CoA, a reaction catalyzed by the enzyme 3-hydroxy-3-methylglutaryl-coenzyme A synthase (HMGCS) [45]. The enzyme 3-hydroxy-3-methylglutaryl-coenzyme A reductase (HMGCR) then converts HMG-CoA into mevalonate [46]. Subsequently, multiple enzymatic reactions produce isoprenoids, which are condensed to produce a squalene molecule. Squalene is then cycled, generating lanosterol molecules that are converted into cholesterol [45]. Finally, cholesterol can be esterified by the enzymes Acyl-CoA: cholesterol O-acyltransferases (ACAT1 and ACAT2), that yield CE which can be incorporated into LDs [47].

Cytoplasmic LDs are produced in ER microdomains, where the synthesis of TAGs, one of their main components, also occurs [48]. Nuclear LDs emerge in the inner nuclear membrane [7]. Two models have been proposed for cytoplasmic LDs and their interaction with the ER [49]. On the one hand, mature LDs emerge from the ER into the cytoplasm with the possibility that the LDs remains attached to the ER by close interactions with its membrane [49]. A second model proposes that LDs remain in the ER and are surrounded by the ER membrane [50]. The intrinsic molecular curvatures of different phospholipids and other lipids present in the ER can ultimately determine whether certain LDs remain anchored or detach from the ER [50]. Lipids with negative intrinsic curvatures, such as DAGs and phosphatidylethanolamine (PE), favor the inclusion of LDs, while lipids with positive intrinsic curvatures, such as lysolipids, promote the emergence of LDs from the ER [50]. Moreover, two subpopulations of LDs can be characterized within the growth phase of LD formation: initial LDs (iLDs) and expanding LDs (eLDs). iLDs are formed from the ER, where neutral lipids dispersed in the ER eventually accumulate and promote oil lens formation. On the other hand, eLDs are specialized iLDs that colocalize with TAG synthesis by the action of glycerol-3-phosphate acyltransferase 4 (GPAT4) and diacylglycerol O-acyltransferase 2 (DGAT2) proteins [50]. When enough TAGs accumulate in the ER bilayer, LDs outbreak into the cytosol where reticulum proteins, such as fat storage-inducing transmembrane 2 (FIT2), perilipins (e.g., perilipin 2 (PLIN2), and perilipin 3 (PLIN3, also known as TIP47), and seipin participate in these processes [50,51,52]. Importantly, the FIT2 protein binds to DAGs and TAGs and has been reported to have a role in partitioning neutral lipids in the ER to form LDs [53]. TIP47, on the other hand, stabilizes early LD formation [54], and seipin is involved in the growth of nascent shoots from iLDs into mature LDs [52]. Moreover, LDs can bud from the ER into the cytoplasm and increase their size through fusion processes among LDs, or through local lipid synthesis.

Several factors participate in the regulation of FA and sterol synthesis, such as the sterol regulatory element-binding protein 1 (SREBP1), the liver X receptor-α (LXR-α), and peroxisome proliferator-activated receptors (PPARs), among others. SREBP1 acts as a transcription factor that binds to sterol regulatory element-1 (SRE1) to promote mevalonate and FA synthesis pathways [55,56]. LXR-α, a member of the nuclear receptor family, controls the expression of cholesterol transport proteins via transcriptional regulation, a process that regulates the activation of SREBP-1 [57]. On the other hand, three isoforms of PPAR proteins have been described, PPARα, PPARβ/δ, and PPARγ [58]. PPARs usually interact with the retinoid X receptor (RXR) to regulate proteins involved in fat metabolism [59]. Moreover, endogenous ligands such as polyunsaturated FA, saturated FA, eicosanoids, leukotrienes, oxidized FA, and oxidized phospholipids modulate PPAR activity, with PPARα regulating FA oxidation pathways, ketogenesis, and lipid storage [60,61]. Moreover, PPARβ/δ plays an essential role in FA and glucose oxidation in the liver, skeletal muscle, and heart [62,63]. PPARγ, in turn, is mainly expressed in white adipose tissue, modulates whole-body lipid metabolism, controls insulin sensitivity, and promotes the transcriptional regulation of perilipins [64,65].

Interestingly, the unsaturated FAs that form LDs in epithelial and certain immune cells are also involved in arachidonic acid (AA) metabolism. Arachidonic acid is a 20-carbon unsaturated FA extracted from membrane phospholipids by the enzymatic activity of phospholipase A2 (PLA2) [66]. Then, the cyclooxygenase domain of prostaglandin-endoperoxide synthase 2 (COX-2) metabolizes AA into prostaglandin G_2_ (PGG_2_) and subsequently, the COX-2 reductase domain reduces PGG_2_ into prostaglandin H_2_ (PGH_2_). The synthesis of PGH_2_ acts as a precursor for several prostaglandins, such as PGE_2_, PGD_2_, and PGI_2_, among others, as well as thromboxanes (e.g., TXA_2_ and TXB_2_), that are involved in numerous cellular processes [67,68]. Studies in different cell types, such as epithelial cells (IEC-6, CACO-2, and EAC cells) and macrophages infected with *Mycobacterium bovis* bacillus Calmette-Guerin (BCG), showed that LDs co-localize with COX-2 [69]. Importantly, this colocalization has been associated with increased prostaglandin E_2_ levels at the sites where LDs were localized [70,71,72,73,74].

### 2.2. Neutral Lipid Degradation

Neutral lipids stored in LDs can be degraded when processed by TAG lipases and CE hydrolases, as well as by lipophagy [75]. These processes generate free fatty acids (FFA) which can be used to obtain energy through β-oxidation as well to obtain cholesterol and phospholipids, which can be used to form membrane components or synthesize steroidal hormones [49]. In addition, during TAG hydrolysis, LD-associated proteins such as perilipins act as activating proteins for lipases [76]. Three main lipases have been described in mammals [77]: 1. Adipose triglyceride lipase (ATGL) catalyzes the hydrolysis reaction of TAGs to generate DAGs and an FFA [76]. 2. The hormone-sensitive protein lipase (HSL) catalyzes the hydrolysis of DAG to MAGs and an FFA [78,79]. HSL is regulated by hormones and activated by a cellular signal that generates the phosphorylation of HSL by the protein kinase A (PKA), producing HSL translocation to LDs and the degradation of TAGs present in them, with the joint action of ATGL [80]. 3. Finally, monoacylglycerol lipase (MGL) binds to MAGs, generating glycerol and a final FFA molecule [81].

On the other hand, lipophagy is the autophagic degradation of LDs and contributes to lipid turnover [82]. This process requires forming a specialized membrane around the LDs called a phagophore, directing the LDs for degradation [83]. Then, the autophagic membrane (autophagosome) is fused with lysosomes resulting in autolysosomes, which leads to cargo degradation [83]. FFAs generated by lipophagy, due to the breakdown of TAGs, fuel β-oxidation in the mitochondria to produce ATP and restore cellular homeostasis [84]. There are several factors that control lipophagy activation, such as starvation, which increases autophagic sequestration of LDs and their degradation by lysosomes [85], or the overexpression of patatin-like phospholipase domain containing 2 (PNPLA2)/ATGL which promotes the extracellular efflux of FAs via lysosomal exocytosis [86]. Moreover, the overexpression of perilipin 2 (PLIN2) protects LDs against lipophagy in the liver, as PLIN2 deficient cells (Plin2^−/−^ mice) display enhanced lipophagy and have TAGs depleted from the liver [87].

### 2.3. Fatty Acid Uptake

Lipid uptake into the cell also contributes to LD accumulation and provides substrates for neutral lipid synthesis. Multiple proteins participate in this process, such as fatty acid transport proteins (FATPs), fatty acid translocase (CD36), fatty acid-binding proteins (FABPs), and caveolin-1, which allow the uptake of FA into the cytoplasm of the cell [88]. Moreover, internal stimuli such as high levels of reactive oxygen species (ROS) and an increase in the expression of hypoxia factors can also favor an increase in FA uptake, contributing to LD formation [89].

To date, six FATPs have been described (FATP1-6), with these proteins contributing to the formation of LDs, namely through long-chain fatty acid (LCFA greater than 14 carbons) import into the cell [90,91]. FATP displays ACS activity, which suggests that FA reuptake is facilitated by the acylation of these imported lipids [92]. FATP1 is involved in the LCFA uptake including their esterification and oxidation, and their expression is regulated by insulin and peroxisome proliferator-activated receptor (PPAR) activators [93]. Regarding FATP2, this protein is involved in the uptake and activation of exogenous LCFAs, with a preference for n-3 unsaturated FA lipids (C18:3 and C22:6) which are preferentially transported into an acyl-CoA pool that is intended for phosphatidylinositol incorporation instead of phosphatidylcholine (PC), as determined by electrospray ionization/mass spectrometry (ESI/MS) [94].

Additionally, the fatty acid translocase CD36, also called the class B scavenger receptor type 2 (SR-B2), is a multifunctional protein that enhances unesterified FA uptake by the cell, such as oleic acid (OA) and palmitate [95]. Seemingly, CD36 enhances FA uptake and their incorporation into TAGs, increasing the rate of intracellular esterification (in the range of minutes), but not their transport across the plasma membrane (in the range of seconds) [96]. On the other hand, oxidized LDL (oxLDL) also binds to CD36 after FA interaction with the CD36 translocase, and cellular cholesterol stimulates the acute uptake of palmitate by CD36 redistribution [97,98]. Regarding FA translocase regulation, AMPK activation allows the recruitment of CD36, resulting in a rise in FA uptake and FA oxidation. In contrast, CD36 expression suppresses AMP-activated protein kinase (AMPK) phosphorylation and thus, its activity when the translocase is not bound to FA [99]. Furthermore, CD36 glycosylation, ubiquitination, and palmitoylation can impact its activity [100]. While phosphorylation of CD36 regulates FA transport, N-linked glycosylation, ubiquitination, and palmitoylation affect CD36 expression and stability without impacting FA uptake [100]. Interestingly, a recent study showed that the class B scavenger receptor type 1 (SR-B1), which is mainly in charge of facilitating the delivery of CE and cholesterol from HDL particles to the liver, as well as modulating cholesterol efflux from the cell, also plays a role in facilitating FA uptake [101].

On the other hand, FABPs are abundant intracellular proteins expressed in almost all tissues with a high-affinity binding capacity for saturated and unsaturated LCFAs [102]. The leading roles related to FABPs include assimilating dietary lipids in the intestine, regulating lipid storage by FA uptake, as well as their transport and oxidation [102]. Moreover, FABPs have been associated with the plasma membrane (FABPpm) and coupling FA uptake with cellular redox shuttles for controlling oxidative metabolism [103]. Cell muscle contraction and AMPK activation cause the upregulation of FABPpm expression and its translocation from the cytoplasm to the plasma membrane [104].

Finally, caveolin-1 (Cav-1) overexpression in adipocytes relates to an increased number of caveolae that enhance the fat accumulation and LD formation through FA uptake [105]. Depletion of Cav-1 showed an increase in TAGs and FAs in blood plasma samples, suggesting a reduction in cellular FA uptake [106,107]. Moreover, a recent study suggests that the ps15-homology domain-containing protein 2 (EHD2), a dynamin-related ATPase located at the neck of caveolae, controls a caveolae-dependent FA uptake pathway [108]. Conversely, Cav-1 is an LD-coating protein that controls LD formation in endothelial cells by negative lipolysis regulation [109]. Studies in endothelial cells lacking Cav-1 showed an impaired accumulation of LDs due to increased lipolysis without reducing TAG synthesis or FA uptake. Furthermore, Cav-1 deletion led to an increase in cAMP/PKA signaling, allowing the phosphorylation of HSL and increased lipolysis [109].

## 3. Modulation of Neutral Lipid Metabolism by Double-Stranded DNA Viruses

Accumulating evidence indicates that the regulation of neutral lipid metabolism by viruses has an important role in their replication cycle and impacts infection [37,110]. The effects of LDs in the life cycle of viruses have been mainly associated with RNA viruses, for which impacts on processes such as viral assembly and overall infection have been reported. Flaviviruses, such as the hepatitis C virus (HCV) and dengue virus (DENV), are RNA viruses extensively studied for how they modulate and are modulated by neutral lipid metabolism and LD regulation during infection [111,112]. For instance, HCV infection is associated with extrahepatic disorders, neutral lipid metabolic alterations, and LDs accumulation, which impact the production of viral particles from infected cells [113,114]. Furthermore, studies with different HCV genotypes indicate that HCV core and non-structural proteins interact with LDs, affecting virus assembling and release to the extracellular medium [111,115]. Moreover, LD accumulation favors the replication of the HCV genome in liver cells (Huh-7 cells) [116]. Additionally, severe acute respiratory syndrome coronavirus-2 (SARS-CoV-2) has emerged as an RNA virus that also modulates lipid metabolism during infection [117]. During SARS-CoV-2 infection, viral particles colocalize with LDs and promote neutral lipid synthesis [117].

Although there is abundant literature on the role of lipid metabolism and LDs over RNA viruses, less has been reported for DNA viruses, which we review and discuss herein.

### 3.1. Adenoviruses

Adenoviruses are non-enveloped viruses that can produce a wide range of diseases in the respiratory, gastrointestinal, and urinary tract [118]. The Adenoviridae family is highly prevalent in the human population, and 67 immunologically different serotypes have been identified [118]. During the last decade, some studies have reported effects on neutral lipid metabolism by different adenovirus serotypes (Table 1) (Figure 2A) [119,120,121,122,123,124,125].

Human adenovirus serotype 2 (Ad2), which mainly infects the respiratory tract, has been reported to produce LD formation in Ad2-infected adenocarcinoma human alveolar basal epithelial (A549) cells, which was mediated by CE accumulation [120]. CE synthesis associated with Ad2 infection was found to be produced by the activation of a complex called oxysterol-binding protein-related protein 1L (ORPL1L)-vesicle-associated membrane protein (VAMP)-associated proteins (VAP), which is in charge of regulating late endosome function under the regulation of Rab7-GTP [119,120]. Importantly, the ORP1L-VAP complex supports the transport of LDL-derived cholesterol from endosomes to the ER, where cholesterol is esterified to CE and later stored in LDs when ORP1L was bound to the ER early region 3 membrane protein RIDα of Ad2 by a RAB7-independent mechanism [120]. In contrast, studies in human adipose-derived stem cells (hADSCs) showed that Ad2 infection did not modulate LD accumulation and FA oxidation in these cells [123,124].

Furthermore, research on adenovirus serotype 5 (Ad5), which is also a common cause of respiratory tract infection in humans, reported that monocytes and PBMCs obtained from Ad5-seropositive subjects accumulated significant levels of neutral lipids (Figure 2A) [118,121]. This neutral lipid accumulation was related to a high accumulation of total cholesterol (TC), LDL, and high-density lipoprotein (HDL) in the serum of Ad5-seropositive subjects, but not by TAG accumulation, due to the fact that TAGs levels measured in monocytes derived from Ad5-positive individuals were lower than that in control monocytes from Ad5-negative persons [121].

Human adenovirus serotype 31 (AdV31), which mainly infects the gastrointestinal tract, has also been associated with adipogenesis induction [118,122]. Studies in a mouse embryonic fibroblast adipose-like cell line (3T3L1) suggest that infection with this virus enhanced pre-adipocyte differentiation into fat cells and upregulated lipid accumulation through the activation of the CCAAT/enhancer-binding protein (C/EBP-b) and PPARγ-related genes [122]. However, no reports have been presented regarding the accumulation of LDs in 3T3L1 cells.

Human adenovirus serotype 36 (Ad-36), which also produces gastrointestinal disease and is highly associated with obesity, has also been reported to modulate lipid metabolism [118,141]. Studies in primary cultured human skeletal muscle cells suggested that Ad-36-induced LDs were associated with high neutral lipid Oil Red O staining (ORO) [123]. Moreover, Ad-36 infection significantly reduced FA oxidation, but, in contrast, increased cell death-inducing DFFA-like effector C(Cidec)/fat-specific protein 27 (FSP27), ACC, sterol regulatory element-binding protein 1c (SREBP-1c), SREBP2, and HMGCR protein expression [123]. Another study regarding the Ad36 infection of hADSCs cells reported that infection might promote FA and TAG synthesis and subsequent LD formation by modulating phosphoinositide 3-kinase (PI3K)/protein kinase B (Akt)/ forkhead box O1 (FoxO1)/ PPARγ signaling pathway [124].

Moreover, additional studies reported that fowl adenovirus serotype 4 (FAdV-4) induces hepatic injury through TAG accumulation that are shaped similar to LDs in the cytoplasm of hepatocytes [125]. Chickens infected with FAdV-4 underwent induction of the LXR-α, PPARγ, and SREBP-1c adipose-related genes [125]. In contrast, FAdV-4 infection generated the downregulation of LDL secretion-related genes, lipid oxidation, and lipid decomposition-related genes [125]. On the other hand, treatment with an LXR-α antagonist (SR9243) decreased the number of LDs present in the cell and the accumulation of lipogenic genes and virus production [125]. In contrast, cell treatment with an LXR-α agonist (T0901317) increased the number of LDs and the expression of lipogenic genes, suggesting that FAdV-4-induced steatosis involves the activation of the LXR-α signaling pathway (Table 2) [125].

Taken together, different adenovirus infections have impacts on the accumulation of neutral lipids and LDs during infection. Nonetheless, lipid metabolism components altered in the cells during infection have been shown to depend on the cell type infected, suggesting that the cell type phenotype significantly determines those neutral lipid metabolism components modulated by the virus. Although several pathways associated with lipid regulation, such as PPAR, LXR-α, and SREBP, are reported to be modulated by these viruses, there are relatively few studies that investigate the specific roles that neutral lipids and LDs have over the replication cycle of adenoviruses. Therefore, further exploration of the function of neutral lipids and LDs in adenovirus infections is expected for a better understanding of the role played by these components and their potential as therapeutic targets for limiting infections with these viruses.

### 3.2. Herpesviruses

Herpesviruses are enveloped viruses that are divided into three subfamilies: *alphaherpesvirinae, betaherpesvirinae,* and *gammaherpesvirinae* (Figure 2) [148]. The human *alphaherpesvirinae* subfamily is composed of herpes simplex virus type 1 (HSV-1) and type 2 (HSV-2), as well as varicella-zoster virus (VZV) [149,150]. The human *betaherpesvirinae* subfamily contains cytomegalovirus (HCMV), human herpesvirus 6 (HHV-6), and human herpesvirus 7 (HHV-7), while human herpes virus 4 and 8, which correspond to Epstein-Barr virus (EBV) and Kaposi’s sarcoma-associated herpesvirus (KSHV), respectively, belong to the *gammaherpesvirinae* subfamily [148].

#### 3.2.1. Alphaherpesviruses

##### Herpes Simplex Virus Type 1

Herpes simplex virus type 1 (HSV-1) was recently reported to induce the formation of LDs in astrocytes during early infection (8 h) (Figure 2B) (Table 1) [126]. This study suggested that LD accumulation was mediated by the activation of the epidermal growth factor receptor (EGFR), because the pharmacological inhibition of EGFR by the AG-1478 drug reduced LDs accumulation, altogether increasing viral replication (Table 2) [126]. Moreover, this study showed that LD accumulation mediated by HSV-1 infection enhanced IFN-β, IFN-λ, and viperin expression, restricting HSV-1 genome replication [126]. Another study in epithelial cells (COS-7) reported that the viral glycoprotein D (gD) interacts with the viperin protein, and both colocalize on the surface of LDs, as well as in the Golgi apparatus [127]. However, this report did not show evidence that HSV-1 induces the formation of LDs in COS-7 cells [127]. In epithelial cells, HSV-1 modulates the synthesis of other lipids that are involved in cell signaling processes, such as phosphoinositides; increasing 4,5-phosphatidylinositol bisphosphate (PIP2) and decreasing 4-phosphatidylinositol (PIP) [151]. On the other hand, there was an increase in the de novo synthesis of phospholipids after HSV-1 infection in epithelial cells, which was determined necessary for producing new viral particles [152,153]. 

##### Varicella-Zoster Virus

Several studies have indicated that the varicella-zoster virus (VZV) can modulate the synthesis of neutral lipids in MRC-5 cells and human embryonic lung cells, mainly TAG synthesis depending on the VZV strains, but not CE accumulation (Table 1) [128]. Another study indicated that VZV glycoproteins display FA acylation, which was necessary for producing newly infectious virions [142]. Interestingly, the use of cerulenin, an antibiotic that inhibits the de novo biosynthesis of FA, revealed a decrease in mature glycoproteins [142]. Moreover, cerulenin significantly inhibited viral growth, but not VZV protein synthesis in human embryo fibroblast cells [142]. Furthermore, the use of orlistat, a drug that inhibits lipoprotein lipase (LPL) and FAS proteins, reduced VZV replication in human embryonic lung fibroblast (HELF) cells (Table 2) [143]. However, at present, it is unknown whether VZV promotes LD accumulation, as well as the regulation of neutral lipid synthesis.

Altogether, there are few reports studying the effects of lipid metabolism on human alphaherpesvirus and viral infection. While infection with HSV-1 has been recently reported to induce LDs in astrocytes and epithelial cells, this remains to be determined for other cell types that are targets for this virus, such as immune cells and other cells within the central nervous system. Additionally, it will be interesting to assess whether the observation that VZV controls TAG synthesis differentially depends on whether the virus is undergoing a lytic or latent phase, and if this also applies to other alphaherpesviruses, namely, HSV-1 and HSV-2 [128]. Interestingly, recent studies with alphaherpesviruses affecting animals, such as Marek’s disease virus (MDV), suggest that these viruses can significantly alter lipid metabolisms processes in infected cells, likely promoting viral fitness and infectious processes [154]. It will be interesting to determine whether these effects also apply to human alphaherpesvirus. Therefore, additional studies are needed to better determine how these latter viruses affect lipid metabolism and LD formation and to eventually harness these findings for potential new therapies against these viruses to counteract them.

#### 3.2.2. Betaherpesviruses

##### Cytomegalovirus

Cytomegalovirus (CMV) is a betaherpesvirus identified as the causal agent of the disease mononucleosis and is also associated with some types of cancers and cardiovascular diseases [155]. CMV has been reported to control lipid metabolism (Figure 2C) (Table 1). After infection with CMV, increases in the neutral lipids amounts in human embryonic lung cells and smooth muscle cells from human saphenous veins have been observed [129]. When separating the content of the neutral lipids within these cells into DAG, cholesterol (C), TAG, and CE, it was found that Ad-169-infected cells had lower CE, TAG, and DAG components and higher cholesterol levels than non-infected cells [129]. Similar results were observed in human fibroblasts infected with CMV, where infection increased the efflux of cellular cholesterol despite reducing the abundance of the ATP-binding cassette transporter 1 (ABCA1) [156]. CMV was reported to increase lipogenesis in infected cells due to a redistribution of the viperin protein to the mitochondria where it interacts with and blocks the function of the mitochondrial trifunctional protein (TFP), which is in charge of mediating β-oxidation of FAs [157]. The inhibition of TFP activity resulted in a decrease in cellular ATP levels and the subsequent activation of AMPK, which promotes the expression of the glucose transporter GLUT4 [157]. High levels of GLUT4 expression increase glucose import, as well as nuclear translocation of the glucose-regulated transcription factor carbohydrate-responsive element-binding protein (ChREBP), which ultimately induces the transcription of genes encoding lipogenic enzymes, resulting in a rise in lipid synthesis, LD accumulation, and viral envelope generation [157]. Moreover, HCMV induces the expression of low-density lipoprotein receptor (LDLR), scavenger receptor class B (SCARB), cholesteryl ester transfer protein (CETP), HMGCR, apolipoprotein B (ApoB), and LPL in human umbilical vein endothelial cells [158]. On the other hand, the accumulation of LDs has also been reported in fibroblasts infected with murine CMV (mCMV) [159]. Studies in vivo have shown that mCMV infection resulted in elevated plasma TAGs and disturbed cell cholesterol metabolism without affecting the degree of functionality of high-density lipoprotein (HDL) [156]. Finally, in vivo studies suggest that HCMV infection plays a role in the pathogenesis of the non-alcoholic fatty liver disease (NAFLD), with liver cells promoting high TAG accumulation and facilitating hepatic steatosis. This disease is promoted by the action of HCMV-encoded IE2, which increases SREBP1c overexpression in the liver [160].

At present, the role of neutral lipid metabolism in human betaherpesvirus infection has only been studied for CMV, and not the human herpesvirus 6 (HHV-6) and human herpesvirus 7 (HHV-7). Interestingly, CMV infection regulates cholesterol synthesis in different cell types, while alterations in TAG synthesis were mainly detected in in vivo studies. Despite the currently available studies regarding lipid regulation in betaherpesvirus infection, it will be necessary to perform additional studies that address possible interactions between viral proteins with LDs or other enzymes involved in neutral lipid synthesis pathways. In addition, a better picture of the different enzymes involved in synthesizing neutral lipids during infection with CMV and other betaherpesviruses could favor the identification of potential new therapeutic targets against CMV infection.

#### 3.2.3. Gammaherpesviruses 

##### Epstein–Barr Virus

Epstein–Barr virus (EBV) is a gammaherpesvirus that causes infectious mononucleosis and is associated with multiple lymphoid and epithelial malignancies. Importantly, EBV drives B-cell proliferation to fuel the pathogenesis of multiple lymphomas [161]. 

Regarding lipid metabolism, EBV immediate-early (IE) protein BRLF1, a transcription factor that promotes the lytic form of EBV infection, activates FAS cellular gene expression through a mechanism based on the activation of the p38 stress mitogen-activated protein kinase in human epithelial tongue cells that are lytically infected with EBV (Table 1) [130]. Moreover, the pharmacological inhibition of FAS activity by cerulenin and C75 drugs prevented BRLF1 activation of BZLF1 EBV protein and early (BMRF1) lytic expression, suggesting that FAS activity is relevant for the induction of BZLF1 transcription from the intact latent genome (Table 2) [130]. Recently, using a proteomic approach, it was identified that human B cells infected with EBV display a high induction of cholesterol and FA biosynthetic pathways, with the Epstein–Barr nuclear antigen 2 (EBNA2), SREBP, and MYC proteins having an essential role in the induction of cholesterol and FA pathways [131].

On the other hand, monocytes infected with EBV display inhibition of prostaglandin E_2_ (PGE_2_) due to the suppression of inducible COX-2 expression at both the transcriptional and translational levels [132]. In contrast, studies in patients with nasopharyngeal carcinoma (NPC) associated with EBV infection report that the expression of EBV oncoprotein latent membrane protein 1 (LMP1) is associated with the induction of COX-2 [162]. Interestingly, the over-expression of COX-2 also produces lytic reactivation of EBV in latently infected cells, a process that is regulated by the COX-2 downstream effector PGE_2_ and prostanoid receptors EP1 and EP4 [163]. Moreover, other roles for LMP1 have also been recently reported, associating this protein with increased de novo lipogenesis [133]. This study showed that LMP1 induces the expression, maturation, and activation of SREBP1, together with the induction of de novo lipid synthesis and LD formation, an effect that did not occur when EBV-infected epithelial cells were treated with a siRNA knocking down LMP1 [133]. Similar results were obtained in EBV-negative Burkitt’s lymphoma (BL), which ectopically expresses LMP1, and in which induction of FAS, and an increase in the formation of LDs, were observed [164]. This study also found that USP2a, a ubiquitin-specific protease, significantly increased LMP1 expression, playing an essential role in stabilizing FAS activity [164].

##### Human Herpesvirus 8: Kaposi’s Sarcoma

The human herpesvirus 8 (HHV8) is a gammaherpesvirus that causes Kaposi’s sarcoma (Kaposi’s sarcoma-associated herpesvirus (KSHV)). This virus enters in a latency period in B-lymphocytic cells and the vascular endothelium and is known to induce the modulation of lipid metabolism [165]. HHV-8 induces LD accumulation in infected endothelial cells, predominantly via TAG synthesis during the lytic phase, while the synthesis of CE increases during the latent phase [134]. Interestingly, CE synthesis impacts neotubule generation due to the inhibition of cholesterol esterification, which reduces neotubule formation mainly in latently infected cells [134]. This suggests that the reprogramming of CE metabolism is involved in the regulation of neoangiogenesis in HHV8-infected cells and probably plays a role in the high metastatic potential of the derived tumors (Figure 2D) (Table 1) [134].

Moreover, COX-2 is also related to KSHV latency and pathogenesis. Through promoter analyses using human COX-2 promoter deletion and mutation, reporter constructs showed that the nuclear factor of activated T cells (NFAT) and the cyclic AMP (cAMP) response element (CRE) modulate KSHV-mediated transcriptional regulation of COX-2/PGE_2_ to establish and maintain latency (Table 1) [135].

Taken together, available data on human gammaherpesviruses evidences an important modulation of neutral lipid synthesis and LD accumulation which seemingly are modified depending on whether the virus is undergoing lytic or latent viral states. Moreover, these studies suggest that LD accumulation mediates inflammatory processes within EBV- and KSHV-infected cells which may shed light on the mechanisms that yield inflammation upon infection with these viruses and potential new targets to reduce this process. It will be interesting to determine if other DNA viruses, particularly herpesviruses, also modulate the lipid metabolism pathways modulated by KHSV and EBV and the impact they have on infection and disease. Again, a better understanding of the role of neutral lipid metabolism within infected cells could favor the identification of novel therapies against these viruses.

### 3.3. Hepadnaviridae

#### Hepatitis B Virus

Hepatitis B virus (HBV) is a virus that mainly causes chronic liver disease, resulting in hepatitis, cirrhosis, and hepatocellular carcinoma damage [166]. The HBV protein X (HBx) has been identified as a regulator of lipid accumulation in hepatic cells through SREBP1, but not PPARγ induction (Table 1) [167]. In contrast, in vivo and in vitro studies suggest that mice that express the HBx protein (HBx-Tg mice) and that HepG2-GFPHBx stable cells display increased SREBP and PPARγ expression [136]. More specifically, HBx interacts with liver LXR-α, promoting LXR-α binding to the LXR-response element (LXR) [136]. The interaction between LXR-α and LXR results in the upregulation of SREBP1, FAS, and PPAR expression, accompanied by LD accumulation and hepatic steatosis disease, with SREBP1 protein levels being increased and localized in the nucleus to act as a transcription factor and subsequently modulate HBV replication [168,169,170,171]. HBx-NF-κB complex activation is also involved in hepatic liver injury, with tumor necrosis factor receptor 1 (TNFR1) and NF-κB-dependent pathways regulating steatosis and apoptosis [136]. Further, depletion of TNFR1 and NF-κB inhibition with the Bay 11-7082 drug resulted in the inhibition of LD accumulation and downregulation of SREBP1c and PPARγ expression [136]. Furthermore, studies report that HBx-expressing cells present mitochondrial dysfunction which produces high levels of mitochondrial ROS that contribute to an increase in LD formation and HBV gene expression [137,138]. In contrast, the removal of ROS through treatment with N-acetylcysteine decreased LD accumulation in a time-dependent manner [137]. Furthermore, studies in patients with HBV chronic infection indicate that serum FAS concentration is significantly high in these patients, which correlates with the degree of liver steatosis [139].

In contrast with the notion that HBV infection of hepatic cells increases LDs accumulation, studies in epithelial cells show a reduction in intracellular TAGs in infected cells that reduces the mean size of LDs [140]. This effect would be associated with the downregulation of proteins that participate in LD expansion, such as CIDEB and CIDEC [140]. However, the elimination of CIDEB or CIDEC significantly reduced the amount of detectable genetic material of HBV, while the expression of these proteins recovered the production of HBV, suggesting that HBV regulates its viral replication through CIDEB and CIDEC [140].

Regarding FA uptake proteins, hepatoma cells infected with HBV express high levels of FABP1, which is positively modulated by the HBx protein [172]. Interestingly, experiments with ectopic HBx resulted in FABP1 upregulation; in contrast, HBx abolishment decreased FABP1 expression and reduced lipid accumulation [172]. On the other hand, CD36 is overexpressed in HBV-infected HepG2.2.15 cells, in which CD36 expression promotes an increase in cytosolic calcium levels that positively impacts HBV replication [173]. Conversely, CD36 knockdown did not promote HBV replication [174]. Interestingly, hepatic FA in HBV infection indicates that palmitate, stearate, and oleate molecules that are incorporated into the intracellular medium by FABP and CD36 increase HBx protein stability by preventing proteasome-dependent degradation [175]. This stabilization process needs ROS production and intracellular Ca^2+^ signal activation, and these results suggest that liver FA levels might affect HBV-induced pathogenesis [175].

Regarding treatments, the pharmacological inhibition of enzymes that catalyze the FA biosynthesis pathways (Table 2), namely, ACC1 and FAS with CP64018 and GSK1995010 in the context of HBV infection of HepG2-2-15-7 cells, caused a decrease in extracellular HBV DNA in a dose-dependent manner, but not at the intracellular level [144]. These results suggest that these drugs did not inhibit HBV genome replication but rather modulated HBV particle production [144]. In contrast, the MK8245 drug, which inhibits stearoyl-CoA desaturase 1 (SCD1), did not induce a decrease in extracellular HBV DNA in infected HepG2.2.15.7 cells [144]. Moreover, an extract of the herbal medicine *Graptopetalum paraguayense* (GP), also called HH-F3, suppressed the HBV core promoter activity and PGC-1α expression, which is a major metabolic regulator of gluconeogenesis and lipogenesis [145]. HH-F3 also inhibited FAS expression and decreased lipid accumulation by PGC-1α. This down-regulation correlated with the inhibition of HBV replication. Therefore, these studies suggest that targeting PGC-1α may be a therapeutic strategy for treating HBV infection and the pathologies associated with HBV infection [145]. Regarding neutral lipid degradation, lipase inhibition using the orlistat induced the inhibition of HBV infection in hepatocytes after virus inoculation [146]. In contrast, post-treatment with orlistat did not affect HBV gene expression or replication, suggesting that the inhibition of cellular hepatic lipase may target the early steps of HBV infection [146]. Another study showed that the inhibition of ACAT negatively impacts HBV replication in human HepG2 [147]. Moreover, the use of the ACAT inhibitor enhances peripheral and intrahepatic HBV-specific CD8^+^ T cell responses with an increase in the secretion of IFN-γ [147].

Studies on the hepatitis B virus and lipid metabolism show significant impacts on the regulation of lipid metabolism and, notably, that pharmacological inhibition of different lipid metabolism pathways has effects on HBV particle production and HBV DNA replication. These findings highlight the different roles that lipid metabolism components may have over distinct steps of the replication cycle of viruses and the importance of dissecting the viral processes that are directly affected by cellular lipid components to determine their interplay. An important challenge in this regard is to determine whether lipids have direct or indirect effects on virus components.

## 4. Conclusions

LD biogenesis is regulated by FA uptake, neutral lipid synthesis, and neutral lipid degradation which play different roles in the cell, including the storage of high energy sources and the regulation of oxidative stress, among others. However, there is accumulating evidence that indicates that these neutral lipid-rich organelles differentially modulate DNA virus infections. Indeed, as reviewed herein, adenoviruses and hepatitis B viruses alter lipid metabolism within infected cells to favor their replication. Contrarily, upon herpesvirus infections, LD accumulation has different impacts on virus replication. Overall, these studies highlight the fact that neutral metabolism plays a relevant role in the replication of DNA viruses through a wide range of mechanisms. Nevertheless, several important challenges remain to be addressed in this field to better understand the interplay between lipid metabolism, LDs, and viral components. Indeed, it will be important to extend the current studies to other cell types, identify key lipid metabolism-related enzymes modulated by viral infection, perform high-resolution lipidomic analyses to identify key functional lipids, as well as determine whether the observed effects of lipids over the replication cycle of viruses and vice-versa are mediated directly or indirectly by viral and host determinants. Given the findings described to date regarding lipid metabolism involvement in DNA virus replication and virus infection over lipid homeostasis in the cell, we foresee upcoming studies that will extend the currently available knowledge in this field and significantly deepen our understanding of this interaction. Importantly, this could help pinpoint potentially new therapeutic targets to control virus replication and their associated pathologies. 

## Figures and Tables

**Figure 1 cells-11-02224-f001:**
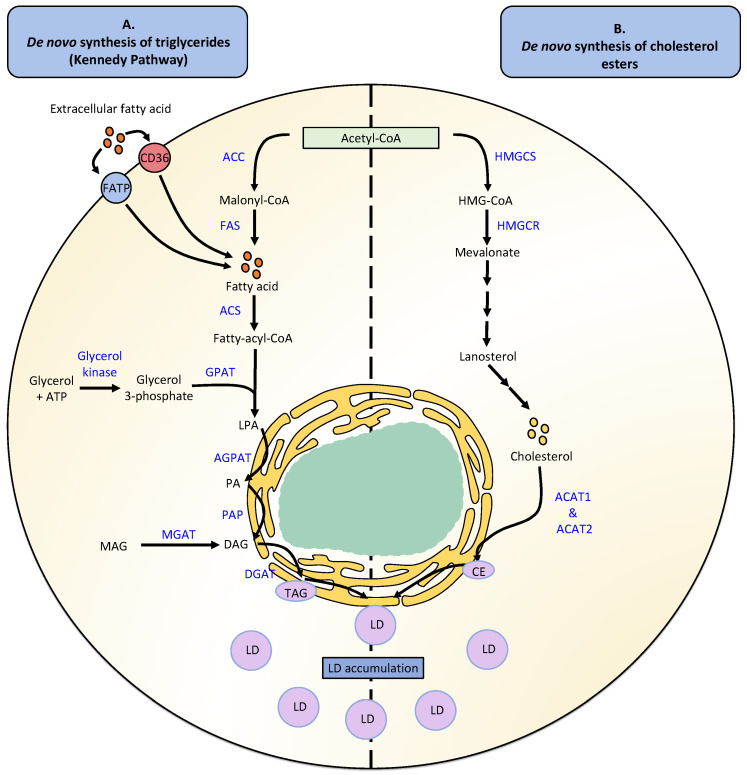
Lipid droplet biogenesis. (**A**) De novo synthesis of triglycerides in a typical human cell resented as a light-yellow circle. (**B**) De novo synthesis of cholesterol esters. Enzyme names are written in blue. Chemical transformations involving multiple reactions are depicted as a sequence of three arrows. Abbreviations: CoA: coenzyme A; HMG: hydroxymethylglutaryl; FAS: fatty acid synthase; ACS: Acyl-CoA synthetase; PA: phosphatidic acid; GPAT: glycerol-3-phosphate O-acyltransferase; LPA: lysophosphatidic acid; AGPAT: 1-acylglycerol-3-phosphate O-acyltransferase; PAP: phosphatidic acid phosphatase; MAG: monoacylglycerol; MGAT: Monoacylglycerol acyltransferase; DGAT: diacylglycerol O-acyltransferases; HMGCS: 3-hydroxy-3-methylglutaryl-coenzyme A synthase; HMGCR: 3-hydroxy-3-methylglutaryl-coenzyme A reductase and ACAT: Acyl-CoA: cholesterol O-acyltransferase.

**Figure 2 cells-11-02224-f002:**
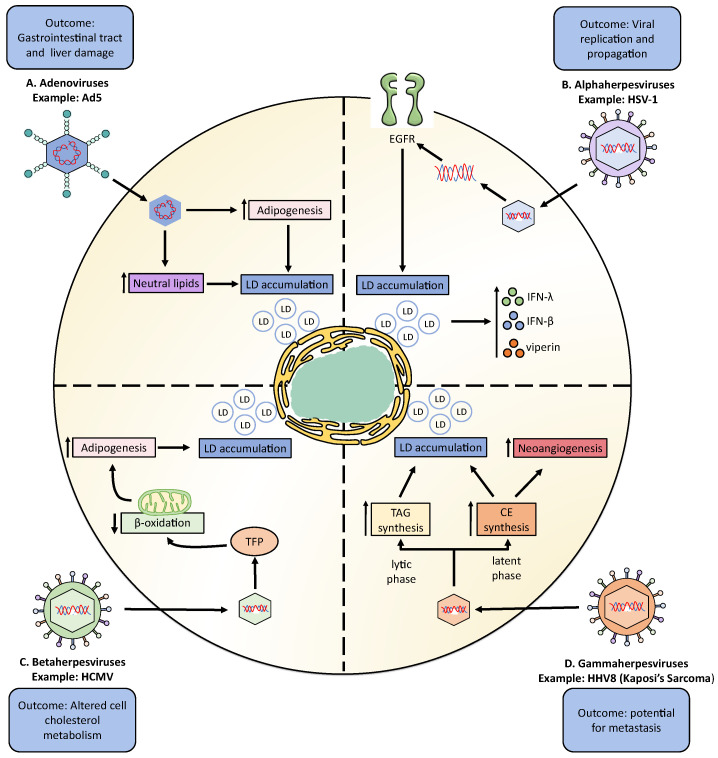
Summary of interactions between selected clinically relevant DNA viruses and LDs leading to pathophysiological outcomes. (**A**) Interactions between Adenovirus 5, an adenovirus, and LDs biogenesis. (**B**) Interactions between herpes simplex virus type 1 (HSV-1), an alphaherpesvirus, and host LDs. (**C**) Interactions between LDs and the human cytomegalovirus (HCMV), a betaherpesvirus. (**D**) Interactions between neutral lipid metabolism and human herpes virus 8 (HHV8), a gammaherpesvirus.

**Table 1 cells-11-02224-t001:** Modulation of lipid metabolism and LD synthesis by DNA viruses.

Virus Family	Virus	LDs Outcome	Cellular Type	Reference
*Adenoviridae*	Human adenovirus serotype 2 (Ad2)	The infection induces LD formation associated with a cholesterol ester accumulation.	Human alveolar basal epithelial (A549) cells	[120]
		The infection does not induce LD accumulation and FA oxidation.	Human adipose-derived stem cells (hADSCs)	[123,124]
	Adenovirus serotype 5 (Ad5)	The infection induces the neutral lipid accumulation associated with high levels of cholesterol, but not TAG.	Human monocytes and PBMCs	[121]
	Human adenovirus serotype 31 (Ad31)	The infection upregulates lipid accumulation by the activation of C/EBP-b and PPARγ genes. It is unknown if the infection regulates LDs accumulation.	Mouse embryonic fibroblast adipose-like cell line (3T3L1)	[122]
	Human adenovirus serotype 36 (Ad36)	The infection induces LD accumulation.	Human skeletal muscle cell	[123]
		Promotes FA and TAG synthesis, producing LD accumulation by the modulation of PI3K/Akt/FoxO1/ PPARγ signaling pathway.	Human adipose-derived stem cells (hADSCs)	[124]
	Fowl adenovirus serotype 4 (FadV-4)	The infection induces TAG accumulation shaped similar to LDs by inducing LXR-α, PPARγ, and SREBP-1c adipose-related genes.	Chicken hepatocytes	[125]
*Herpesviridae*	Herpes simplex virus type 1 (HSV-1)	The infection induces LD accumulation at early times of infection by activating the epidermal growth factor receptor (EGFR).	Astrocytes and HeLa cells	[126]
		Glycoprotein D (gD) interacts with LDs through their interaction with viperin protein.	COS-7 cells	[127]
	Varicella-zoster virus (VZV)	The infection induces the synthesis of TAG, but not cholesterol ester.	Human embryonic lung cells	[128]
	Cytomegalovirus (CMV)	The infection increases neutral lipids synthesis by the accumulation of cholesterol.	Human embryonic lung cells, human saphenous veins smooth muscle cells, and human fibroblast	[129]
	Epstein-Barr virus (EBV)	The viral protein BRLF1 activates the FAS expression.	Human epithelial tongue cells	[130]
		The infection induces the cholesterol and FA synthesis pathways, a process modulated by the EBNA2, SREBP, and MYP proteins.	Human B cells	[131]
		The infection inhibits PGE_2_ expression due to the downregulation of COX-2.	Monocytes	[132]
		The viral LMP1 protein induces de novo lipid synthesis and LD formation.	Epithelial cells	[133]
	Human herpesvirus 8: Kaposi’sarcoma	The infection induces the accumulation of LDs. A TAG accumulation in the lytic phase of infection was produced, meaning in the latent phase of infection, CE are increased.	Endothelial cells	[134]
		The infection increases the expression of COX-2/PGE_2_ signaling.	mECK36 cells	[135]
*Hepadnaviridae*	Hepatitis B virus (HBV)	HBx viral protein induces SREBP, PPARγ, and FAS expression by the interaction of HBx with LXR-α.	HepG2 cells	[136]
		The infection induces LD formation.	HBx-expressing cells	[137,138]
		The infection increases the FAS concentration in HBV chronic patients.	Human serum	[139]
		The infection reduces intracellular TAG accumulation, reducing LD size.	Epithelial cells	[140]

**Table 2 cells-11-02224-t002:** Effects of lipid pharmacological modulation in the replicative cycles of viral infections.

Virus	Drug	Drug Target	Viral Outcome	Cellular Type	Reference
Fowl adenovirus serotype 4 (FAdV-4)	SR9243	LXR-α antagonist	Decreases the virus production and the LD number.	Hepatocytes.	[125]
Herpes simplex virus type 1 (HSV-1)	AG-1478	Inhibitor of epidermal growth factor receptor	Increases viral replication and decreases LD accumulation.	Astrocytes	[126]
Varicella-zoster virus (VZV)	Cerulenin	FAS inhibitor	Inhibits viral growth without affecting VZV protein synthesis.	Human embryo fibroblast cells	[142]
	Orlistat	Inhibitor of LPL and FAS	Inhibits VZV replication.	Human embryonic lung fibroblast cells (HELF)	[143]
Epstein-Barr virus (EBV)	Cerulenin	FAS inhibitor	Inhibits the activity of BRLF1 viral protein and BMRF1 lytic expression.	Human epithelial tongue cells	[130]
	C75	FAS inhibitor	Inhibits the activity of BRLF1 viral protein and BMRF1 lytic expression.	Human epithelial tongue cells	[130]
Hepatitis B virus (HBV)	CP64018	ACC1 inhibitor	Decreases extracellular HBV DNA.	HepG2-2-15-7 cells	[144]
	GSK1995010	FAS inhibitor	Decreases extracellular HBV DNA.	HepG2-2-15-7 cells.	[144]
	MK8245	SCD1 inhibitor	No effect.	HepG2-2-15-7 cells.	[144]
	*Graptopelatum paraguayense* (HH-F3)	Inhibit PGC-1α and FASN expression	Inhibits HBV core promoter activity.	Huh-7, HepG2, and Hep3B/T2 cells	[145]
	Orlistat	LPL and FAS inhibitor	Does not affect HBV gene expression.	Differentiated HepaRG cells	[146]
	Avasimibe	ACAT inhibitor	Inhibits HBV replication.	HepG2 cells	[147]

## Data Availability

Data sharing does not apply to this article as no new data were created or analyzed in this review.

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
