# Peer review of "Interplay between Lipid Metabolism, Lipid Droplets, and DNA Virus Infections"

_cells, 2022, doi:10.3390/cells11142224_

Round 1

Reviewer 1 Report

The authors do an extensive work bringing to the reader several aspects of the lipid metabolism and biology of lipid droplets and their role during DNA virus infection. However, Faria et al. take few opportunities to write their critical thoughts about the literature findings. As the literature on the role of LDs in DNA virus infections is scarce, I believe a crucial point of this work is the lack of comparison of DNA viruses with RNA viruses. The review would gain additional interest and significance if the authors discuss some relevant issues more problem-oriented, bringing up different views and clearly expressing their own critical point of view. 

Major points:

1. The role of LDs in DNA virus infection is much less explored in the literature than in RNA viruses. In this context, I believe that a negative aspect of this article is the lack of comparison between what happens in RNA viruses, particularly on the functions of LDs in viral morphogenesis. 

2. The text also lacks a description of the steps of lipid metabolism that happen at the LDs level. This detailing is essential to clarify what biological processes would happen at the LD and ER levels. 

3.  Despite the very detailed description of the lipid synthesis pathways, the text lacks the central signalling and cellular factors involved in the biogenesis of LDs. Although the authors associate the increased expression of SREBP1, PPAR, and LXR with lipid metabolism in DNA virus infection, there is no sentence in the text about the function of these factors or their involvement with lipid metabolism.

4. As a precursor article, I believe the paper would benefit from being less descriptive and more critical. Based on what we know about DNA viruses, would LDs be more pro-pathogenic or pro-host? What could be the roles of LD in the morphogenesis of DNA viruses?

Minor

1. In the last four years, there have been many new revisions involving LD in host-pathogen interaction (PMID: 34925055, PMID: 31121077, PMID: 33512504), I strongly recommend that the introduction be updated. In addition, I recommend that, whenever possible, the original references of the work are cited. 

2. Replacement of LD by Lipid metabolism and LDs in the title of Table 1 have more metabolism events than LDs.

3. Line 77: Plin2 and Plin3 are the principle LD-structural proteins in the host-pathogens infection - include the plin2 in this sentence.

4. On line 381: the Authors correlate COX2 with the LDs. However, there is no sentence in the text about the function of LD in the eicosanoid synthesis.

Author Response

Reviewer 1:

The authors do extensive work bringing to the reader several aspects of the lipid metabolism and biology of lipid droplets and their role during DNA virus infection. However, Farías et al. take few opportunities to write their critical thoughts about the literature findings. As the literature on the role of LDs in DNA virus infections is scarce, I believe a crucial point of this work is the lack of comparison of DNA viruses with RNA viruses. The review would gain additional interest and significance if the authors discuss some relevant issues more problem-oriented, bringing up different views and clearly expressing their own critical point of view. 

Major points:

1.The role of LDs in DNA virus infection is much less explored in the literature than in RNA viruses. In this context, I believe that a negative aspect of this article is the lack of comparison between what happens in RNA viruses, particularly on the functions of LDs in viral morphogenesis.

Response: We appreciate the Reviewer's comment. A comprehensive review on the role of LDs in DNA and RNA viruses would certainly be of interest to the readers, however, given the current (quite long) extension of the review in its present form, which is focused on the role of LDs in DNA viruses, we have incorporated to the updated version of the manuscript a brief section of the main RNA viruses that interact with LDs.

  1. The text also lacks a description of the steps of lipid metabolism that happen at the LDs level. This detailing is essential to clarify what biological processes would happen at the LD and ER levels.

Response: We appreciate the Reviewer's comment. To address this suggestion, we have updated the manuscript to incorporate a description of the lipid metabolism steps at the LD level. 

  1. Despite the very detailed description of the lipid synthesis pathways, the text lacks the central signalling and cellular factors involved in the biogenesis of LDs. Although the authors associate the increased expression of SREBP1, PPAR, and LXR with lipid metabolism in DNA virus infection, there is no sentence in the text about the function of these factors or their involvement with lipid metabolism.

Response: We thank the Reviewer for this comment. Based on this comment, we have modified the manuscript to incorporate a paragraph that includes the role of SREBP1, PPAR and LXR in lipid metabolism. 

  1. As a precursor article, I believe the paper would benefit from being less descriptive and more critical. Based on what we know about DNA viruses, would LDs be more pro-pathogenic or pro-host? What could be the roles of LD in the morphogenesis of DNA viruses?

Response: We appreciate the Reviewer's comment. Accordingly, based on the suggestion, we have edited the text to incorporate in the manuscript some questions regarding the potential role of LDs over DNA virus infections. 

Minor

1.In the last four years, there have been many new revisions involving LD in host-pathogen interaction (PMID: 34925055, PMID: 31121077, PMID: 33512504), I strongly recommend that the introduction be updated. In addition, I recommend that, whenever possible, the original references of the work are cited. 

Response: We thank the Reviewer for this comment. We have updated the manuscript to include the references suggested by the Reviewer and have reduced the number of cited reviews by including in their place the corresponding original papers.

  1. Replacement of LD by Lipid metabolism and LDs in the title of Table 1 have more metabolism events than LDs.

Response: As requested by the Reviewer, we have made the corresponding modification in the updated version of the manuscript.

  1. Line 77: Plin2 and Plin3 are the principle LD-structural proteins in the host-pathogens infection - include the plin2 in this sentence.

Response: As requested by the Reviewer, we have made the corresponding modification in the updated version of the manuscript.

  1. On line 381: the Authors correlate COX2 with the LDs. However, there is no sentence in the text about the function of LD in the eicosanoid synthesis.

Response: Thank you for this comment. To address this point, we have incorporated to the revised version of the manuscript a paragraph that indicates the role of COX-2, and its relationship with lipid metabolism and LDs regarding eicosanoid synthesis. 

Once again, we would like to thank the Reviewer for the time and effort invested in revising our manuscript. We feel that the manuscript has significantly improved after this revision and hope that it is now suitable for publication in this journal.

Reviewer 2 Report

Farías and colleagues have submitted a review manuscript describing the Interplay between Lipid Metabolism, Lipid Drop-Lets, and DNA Virus Infections. I think this review is a good summary of the recent data in the area of LD accumulation and brings together diverse knowledge of LDs, DNA viruses, and the relationship between the two. This review for the most part is well written, however, I have some minor concerns which I have listed below;

Minor concerns:

1.       Line 57; spelling mistake of “these”

2.       Line 82- this numbered heading is the same as the one before. (2.1 lipid droplet formation)

3.       Line 98- Full name of CE needs to be written for the first time

4.       I think the review would benefit from a detailed figure of the structure of LDs and where the different lipids synthesised are located in a LD. There is a lot of information in the text, however, it is easy to get lost without a figure to summarise all of this.

5.       Line 195- I think the start of this section could benefit from a general summary of what is known about viral infections (in general) inducing LD accumulation. There are some great reviews that you could reference (“Lipid droplets and the host–pathogen dynamic: FATal attraction? “or “Lipid droplets and lipid mediators in viral infection and immunity”). I think it would set the scene about what is currently known in general and allow the transition into DNA viruses specifically.

6.       Line 273- Do viperin and HSV-1 co-localise “in” LDs or just at the surface?

7.       The viral sections lack a summary under each virus family suggesting a role for LDs during that particular infection. I think the addition of this will allow the reader to join the dots that have been made, allowing the article to flow a lot better.

8.       Line 284- I’m not sure this sentence makes sense do you mean that TAG synthesis is modified?

9.        Figure 1- I think this figure is a great addition to the manuscript

10.   When speaking about the neutral lipids in the virus sections, do you mean TAGs? Is there a specific lipid species that you mean, or just general “neutral lipids”? I think because the review has a metabolism slant, it would be good to be more specific about the actual lipid species that are changing in these infections.

11.    Line 351- you mention PGE2, I think this needs a descriptor.

12.   Line 389- I’m not sure why you are starting this sentence with “on the other hand”, am I missing some of the text?

13.   Line 456- in your conclusion you suggest that “lipid-rich organelles [LDs] modulate DNA virus infections” however there is evidence in your text that suggests the opposite- for example reference 95 describes that when LDs are inhibited from the inhibition of EGFR you get less viral replication (ie less LDs= more virus replication). This sentence, and the statement after this needs to be modified, to incorporate all reviewed literature in this review. This is a newer addition to the literature, so it could be the fact that the thoughts on this topic are changing thus making an excellent point to discuss in the conclusion. 

Author Response

Reviewer 2:

Farías and colleagues have submitted a review manuscript describing the Interplay between Lipid Metabolism, Lipid Drop-Lets, and DNA Virus Infections. I think this review is a good summary of the recent data in the area of LD accumulation and brings together diverse knowledge of LDs, DNA viruses, and the relationship between the two. This review, for the most part, is well written, however, I have some minor concerns which I have listed below;

Minor concerns:

1.Line 57; spelling mistake of "these"

Response: As requested, this typo has been amended (line 63).

2.Line 82- this numbered heading is the same as the one before. (2.1 lipid droplet formation)

Response: We thank the Reviewer for noticing this. This has been amended in the updated version of the manuscript (line 67).

  1. Line 98- Full name of CE needs to be written for the first time

Response: As requested by the Reviewer, we have made the corresponding modification in the updated version of the manuscript. We have added the full name of CE (cholesterol ester) when mentioned for the first time (line 26).

  1. I think the review would benefit from a detailed figure of the structure of LDs and where the different lipids synthesized are located in a LD. There is a lot of information in the text, however, it is easy to get lost without a figure to summarise all of this.

Response: We thanked the Reviewer for this comment. We agree with the suggestion and thus, we created a new figure regarding the synthesis of neutral lipids and the structure of LDs, which was added to the revised version of the manuscript. 

  1. Line 195- I think the start of this section could benefit from a general summary of what is known about viral infections (in general) inducing LD accumulation. There are some great reviews that you could reference ("Lipid droplets and the host–pathogen dynamic: FATal attraction? "or "Lipid droplets and lipid mediators in viral infection and immunity"). I think it would set the scene about what is currently known in general and allow the transition into DNA viruses specifically.

Response: We appreciate the Reviewer's comment. Based on this suggestion, we incorporated a brief section of the main RNA viruses that interact with LDs before entering the section regarding DNA viruses and their role in neutral lipid metabolism. The reviews mentioned by the Reviewer were also included in the revised version of the manuscript.

  1. Line 273- Do viperin and HSV-1 co-localize "in" LDs or just at the surface?

Response: Thank you for this comment. The text has been edited to clarify that viperin and HSV-1 glycoprotein D (gD) co-localize at the surface of LDs.

  1. The viral sections lack a summary under each virus family suggesting a role for LDs during that particular infection. I think the addition of this will allow the reader to join the dots that have been made, allowing the article to flow a lot better.

Response: We appreciate the Reviewer's comment. As suggested, we have edited the text to recapitulate at the end of each section the effects of LDs over the different types of DNA viruses.

  1. Line 284- I'm not sure this sentence makes sense do you mean that TAG synthesis is modified?

Response: For better clarity, the indicated sentence was rewritten to better transmit the corresponding notion.

  1. Figure 1- I think this figure is a great addition to the manuscript

Response: We thank the Reviewer for this comment.

  1. When speaking about the neutral lipids in the virus sections, do you mean TAGs? Is there a specific lipid species that you mean, or just general "neutral lipids"? I think because the review has a metabolism slant, it would be good to be more specific about the actual lipid species that are changing in these infections.

Response: We appreciate the Reviewer's comment. The reports referenced at this section refer mainly to TAGs, which is now indicated in the revised version of the manuscript.

  1. Line 351- you mention PGE2, I think this needs a descriptor.

Response: As requested by the Reviewer, we have made the corresponding modification in the updated version of the manuscript to include the corresponding descriptor of PGE2 (prostaglandin E2—line 479). A brief sentence regarding its role in lipid metabolism was also added to the manuscript (lines 147-153). 

  1. Line 389- I'm not sure why you are starting this sentence with "on the other hand", am I missing some of the text?

Response: As suggested by the Reviewer, this sentence was edited.

  1. Line 456- in your conclusion you suggest that “lipid-rich organelles [LDs] modulate DNA virus infections” however there is evidence in your text that suggests the opposite- for example reference 95 describes that when LDs are inhibited from the inhibition of EGFR you get less viral replication (ie less LDs= more virus replication). This sentence, and the statement after this needs to be modified, to incorporate all reviewed literature in this review. This is a newer addition to the literature, so it could be the fact that the thoughts on this topic are changing thus making an excellent point to discuss in the conclusion. 

Response: We appreciate this comment. As suggested, this sentence has been edited. Also, we now include in the Conclusion section a discussion on the role of LDs in different DNA viral infections.

We would like to thank once more the time and effort invested by the editor and Reviewers in revising our manuscript and feel that the changes have significantly improved the text.

Round 2

Reviewer 1 Report

Clearly, the authors have made considerable effort to address the concerns raised by the reviewers and the paper has been improved.  I believe that this manuscript would be a valuable addition to the journal